# Development of a Multihole Atmospheric Plasma Jet for Growth Rate Enhancement of Broccoli Seeds

**Khattiya Srakaew [1], Artit Chingsungnoen [1,*], Waraporn Sutthisa [2], Anthika Lakhonchai [1], Phitsanu Poolcharuansin [1], Poramate Chunpeng [1], Catleya Rojviriya [3], Kanjana Thumanu [3] and Sarayut Tunmee [3]**

[1] Technological Plasma Research Unit, Department of Physics, Faculty of Science, Mahasarakham University, Maha Sarakham 44150, Thailand; khattiya.ball@gmail.com (K.S.); l.anthika26@gmail.com (A.L.); phitsanu.p@msu.ac.th (P.P.); poramat.c@msu.ac.th (P.C.)

[2] Department of Biology, Faculty of Science, Mahasarakham University, Maha Sarakham 44150, Thailand; waraporn.s@msu.ac.th

[3] Synchrotron Light Research Institute (Public Organization), Nakhon Ratchasima 30000, Thailand; catleya@slri.or.th (C.R.); kanjanat@slri.or.th (K.T.); sarayut@slri.or.th (S.T.)

* Correspondence: artit.c@msu.ac.th; Tel.: +66-8971-10157

**Abstract:** This work aims to develop a multihole atmospheric pressure plasma jet (APPJ) device to increase the plasma area and apply it to a continuous seed treatment system. Broccoli seed was used to study the effects of an atmospheric pressure plasma jet on seed germination and growth rate. An argon flow rate of 4.2 lpm, a plasma power of 412 W, and discharge frequency of 76 kHz were used for seed treatment. The contact angle decreased strongly with the increase in treatment time from 20 s to 80 s. The broccoli seed's outer surface morphology seemed to have been slightly modified to a smoother surface by the plasma treatment during the treatment time of 80 s. However, the cross-sectional images resulted from Synchrotron radiation X-ray tomographic microscopy (SRXTM) confirmed no significant difference between seeds untreated and treated by plasma for 80 s. This result indicates that plasma does not affect the bulk characteristics of the seed but does provide delicate changes to the top thin layer on the seed surface. After seven days of cultivation, the seed treated by plasma for 30 s achieved the highest germination and yield.

**Keywords:** atmospheric pressure plasma jet; surface treatment; growth rate enhancement

## 1. Introduction

Under laboratory conditions, plasmas are generated by applying a voltage between two electrodes. At sufficiently high power, the ionized gases consist of equal concentrations of positive and negative charges and many neutral species. In general, plasmas can be classified according to temperature into thermal and nonthermal plasmas, which are also termed cold plasmas [1]. Because it can operate at low temperatures, surface treatment with cold plasma has been used in numerous industries worldwide, such as semiconductor technology, medicine and cosmetics, packaging technology, textiles, and agriculture [2–4]. In atmospheric pressure cold plasma, ion temperature is close to room temperature. In contrast, the electron temperatures can easily be of the order of several eV (1 eV $\cong$ 11,600 K). This electron temperature range (<10 eV) is responsible for rotational and vibrational excitations of molecules [5]. However, the small fraction of tail electrons in the electron energy distribution function (EEDF) with energies of the order 10 eV or even higher can generate many different chemical processes [6]. For example, the steady-state density of radicals in a nitrogen plasma jet (the mole fraction of water molecules in nitrogen gas is 0.01) with an electron temperature of 1 eV have been calculated by Uhm [7]. The results show that most reactive nitrogen species have a density of around $10^{14}$–$10^{16}$ molecules/cm$^3$ [7]. Argon plasma is frequently used for physical process treatment due to an effective energy transfer to the solid surface. The argon ions bombarding the surface can dislodge contamination

from it and roughen it on an atomic scale [8]. Oxygen, nitrogen, or humid air plasmas can be used for surface activation via the gas-phase radicals. When these plasmas are exposed to the surface, different functional groups are created modifying the chemical activity of the surface [9]. The reactive oxygen species (ROS) such as $O_2^-$, $OH^-$, and $O_3$, reactive nitrogen species (RNS) such as NO, $NO_2$, and $NO_3$, and other reactive species are important in retaining the quality of seeds and food products [10–12]. These reactive radical species, especially NO, and also ultraviolet radiation can penetrate into the capsule of seeds and decompose the inner nutriment. This effect can accelerate the activities of the root of the seedling and increase seed germination [10,13].

Two methods have been frequently used to generate atmospheric pressure plasmas, an atmospheric pressure plasma jet (APPJ) and dielectric barrier discharge (DBD). These methods have unique features that are suitable for specific applications [11]. The DBD device consists of two plane-parallel metal electrodes and a dielectric layer covers at least one of these electrodes. The gap which separates the electrodes is limited to a few millimeters wide to ensure stable plasma operation [14]. The APPJ device consists of two concentric electrodes through which the working gas flows. By applying alternating current (AC) power to the inner electrode at a voltage high enough, the gas discharge is ignited [15]. Usually, these techniques are combined for generating and stabilizing atmospheric pressure plasmas. When APPJ is used for large-area processing, multiple jets or multihole arrays with a scanning stage are typically essential for continuous roll-to-roll processes [16]. This technique makes the cold APPs treatment a reliable method to improve seed performance and crop yield [17,18]. Seeds are the most basic and significant means of agricultural production, and high-quality seeds can rapidly germinate and grow [19,20].

Recently, the modification of surface properties of seeds by cold atmospheric pressure plasma treatments has been proposed as a helpful technique to improve seed germination [21–25]. Broccoli seed was chosen as the model for the operational testing of a multihole APPJ device. The effects of seed germination and the growth rate of sprouts on the treatment time were studied. Therefore, this work aims to develop the multihole APPJ device to increase the plasma area and apply it to a continuous seed treatment system.

## 2. Materials and Methods

The experimental set-up is schematically shown in Figure 1. It consisted of a multihole APPJ device with a computer controller for positioning, used for seed treatment, and an alternating current power supply with a discharge voltage range of 0–10 kV and a frequency range of 50–200 kHz used to sustain the plasma. This multihole APPJ device was designed and developed from a laboratory prototype [26]. Argon was used as a carrier gas and injected into the multihole APPJ device with a gas-flow rate range of 1.7–4.2 L per minute (lpm). A high voltage probe (Keysight N2771B, Santa Rosa, CA, USA) with 30 $kV_{peak}$ was used to measure the variable voltages' waveforms at output points of the circuit. The clamp meter current (Pearson 4100, Palo Alto, CA, United States) was used to record the current supplied to the plasma source. The current-voltage waveform was recorded using a two-channel oscilloscope (Agilent technologies DSO1002A, Beijing, China).

The work of adhesion, surface morphology, and cross-sectional images of untreated and treated seeds were examined using contact angle measurement, scanning electron microscopy, and Synchrotron radiation X-ray tomographic microscopy. The SRXTM technique uses X-rays to create cross-sections of a physical object, obtaining three-dimensional (3D) images of samples. In this study, XTM measurement was performed at the end-station of beamline 1.2 W in the Synchrotron Light Research Institute (Public Organization), Nakhon Ratchasima, Thailand. The beamline photon source covered an energy range of 5 to 15 keV. The synchrotron radiation source at the storage ring was generated using a beam energy of 1.2 GeV. The sample was exposed to an incident X-ray beam and rotated through 180° to achieve several projections. These projections were reconstructed to create two-dimensional (2D) slices of the measured volume. The slices could be stacked to recreate the 3D image of

the sample. Broccoli seed was scanned using the XTM technique before and after plasma treatment for comparison of the external surface and internal structure of the seeds.

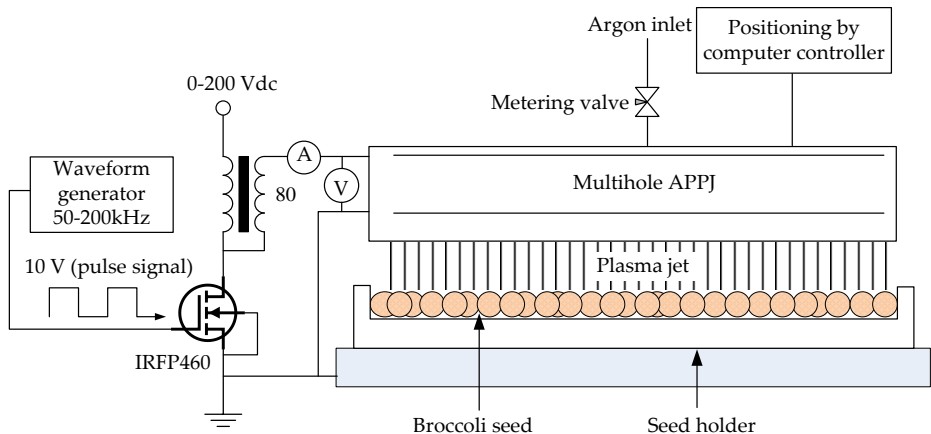

**Figure 1.** Schematic diagram of the experimental apparatus.

To ensure that the plasma beam temperature did not exceed the limit of seed growth, the temperature of the plasma ejected from the nozzle of multihole APPJ must be measured. The experimental set-up of the plasma temperature measurement and the basic diagram of multihole APPJ are schematically shown in Figure 2. The dielectric plate was designed to have five linear arrays with 21 tubes per array on both sides. It was sandwiched with the power and ground electrodes. After plasma was generated with suitable conditions of argon flow rate and discharge power, the plasma beam was ejected from the nozzle with a maximum length of around 10 mm. The spacing between the thermocouple probe and the nozzle was held at 5 mm. A dielectric film was used to cover the thermocouple's probe tip to prevent the built-up charge during the measurement.

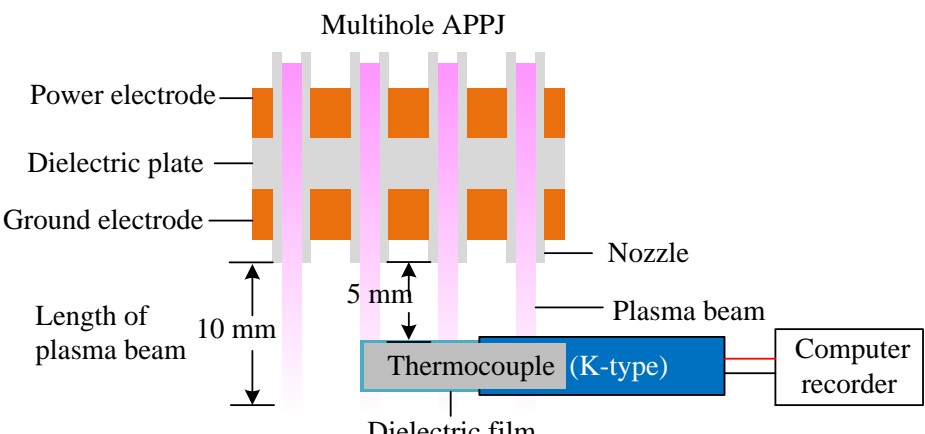

**Figure 2.** Schematic diagram of the multihole plasma jet with temperature measurement (not to scale).

Broccoli seeds with and without plasma treatment were tested for germination ability under laboratory conditions. Two layers of filter paper were soaked in distilled water before placing them in plastic germination boxes, and then 100 seeds of each treatment were added. The germination boxes were incubated at 20 °C for seven days, and the number of germinated seeds was recorded every day. There were three replications in each treatment. Broccoli seeds that had not been plasma-treated were used as a control.

Germination was considered to have occurred when the radicals were half of the seed length. The germination percentage was calculated as follows:

$$\text{Germination percentage (\%)} = \frac{\text{number of germinated seeds}}{\text{total number of seeds}} \times 100, \quad (1)$$

Seedling growth measurements were of shoot length and root length assessed every day for seven days. One hundred broccoli seeds were sown in sterilized peat moss in a seedling tray and watered with 50 mL of distilled water every day for seven days. The experiment was designed as a completely randomized design (CRD) with three replications. After seven days, broccoli sprouts were harvested, and the fresh weight was measured.

## 3. Results and Discussion

### 3.1. Plasma Temperature and Current-Voltage Measurements

Temperature is a crucial factor influencing the germination of seeds [27]. The germination rate increases with rises in temperature up to an optimum value and declines at temperatures exceeding it [28,29]. For most plants, the optimum and maximum germination temperatures are 15–30 °C and 30–40 °C, respectively [30]. High temperatures reduce enzyme efficiency, and eventually, a temperature is reached at which cellular protein is denatured, and the seed is killed [31]. Plasma consists of electrons and ions, which can bombard seed coats, increase the temperature, and affect the germination rate of the seed [32]. Therefore, before seed treatment, it must be ensured that the temperature of plasma does not exceed the upper temperature limits for germination. In this work, the temperature of plasma was measured as a function of exposure time with the sampling rate of 10 Hz, as shown in Figure 3. It found that during 20 min of plasma being exposed to the probe, the maximum temperature was around 36 °C. The temperature did not exceed 27 °C in a 1-min plasma treatment. This result means that the plasma source can be used to treat the seed plant without deteriorated seed populations [24].

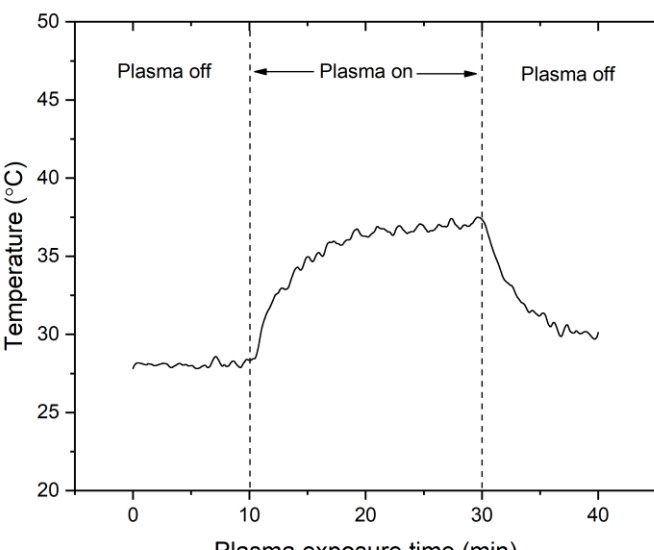

**Figure 3.** The temperature on the probe as a function of the plasma exposure time.

The current and voltage waveforms during the process of plasma treatment are shown in Figure 4. The phase difference between the voltage and the current was 101°. The impedance was practically capacitive, corresponding to the voltage waveform lags behind the current waveform [33]. The plasma can be easily generated, as seen in Figure 5, by using the frequency of 76 kHz. This frequency is suitable to transfer the electrical energy to the plasma source. The root mean squares of voltage $V_{rms}$ of 4.6 kV and current $I_{rms}$ of 410 mA were observed corresponding to the discharge power of 412 W.

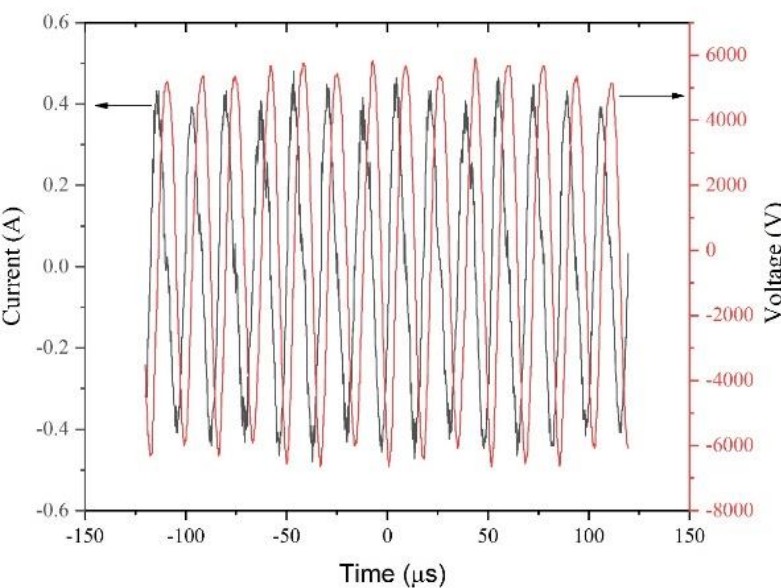

**Figure 4.** Current-voltage waveforms of the multihole plasma jet during the discharge power and frequency of 412 W and 76 kHz, respectively.

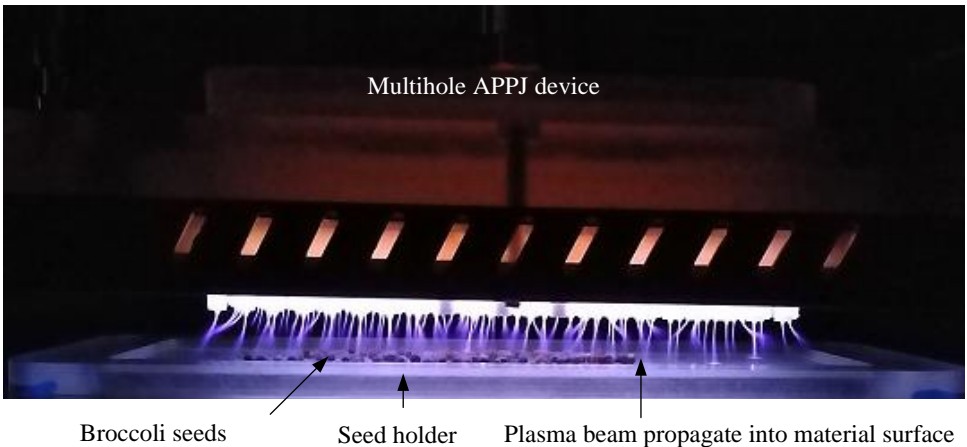

**Figure 5.** Photograph during the multihole plasma jet exposure of the broccoli seeds.

### 3.2. Contact Angle Measurement

Contact angle measurements were carried out using deionized water to determine the broccoli surface's hydrophilicity after being treated by the plasma. The results concerning the effect of multihole APPJ treatment on water wettability are shown in Figure 6. Because the broccoli seeds are small in size and naturally non-uniform in shape and surface structure, the contact angle definition should be modified using an arc surface [34]. As seen in Figure 6a, the contact angle of a liquid with a surface as the mechanical equilibrium of a drop resting on a plane solid surface is the angle between the surface tensions at the interface of the liquid and vapor phases ($\gamma_{lv}$) and at the interface of the solid and liquid phases ($\gamma_{sl}$) [35]. Figure 6b–f shows contact angle images of water droplets adhered to broccoli seeds (b) untreated and (c–f) plasma-treated with treatment times of 20 s, 40 s, 60 s, and 80 s, respectively. For each condition, the mean contact angle was measured using five broccoli seeds. The contact angle and work of adhesion as a function of the treatment time are shown in Figure 7. The contact angle was 130.8 ± 5.5° for the untreated seeds, and the contact angle decreased to 76.4 ± 7.1° when increasing the plasma treatment time. The work of adhesion of a liquid and solid can be calculated directly from the surface tension between liquid and vapor phases and the contact angle [36,37]. A decreasing contact angle

corresponds to increased work of adhesion from $26.0 \pm 1.1$ to $92.5 \pm 8.6$ mN/m. The contact angle decreased strongly, which indicates that the atmospheric pressure argon plasma treatment resulted in dramatic hydrophilization of seeds. The multihole APPJ in the air can produce reactive oxygen and nitrogen species, including $NO_x$, OH, O, and $O_3$. These reactive species stimulate the activation processes of the surface [38,39].

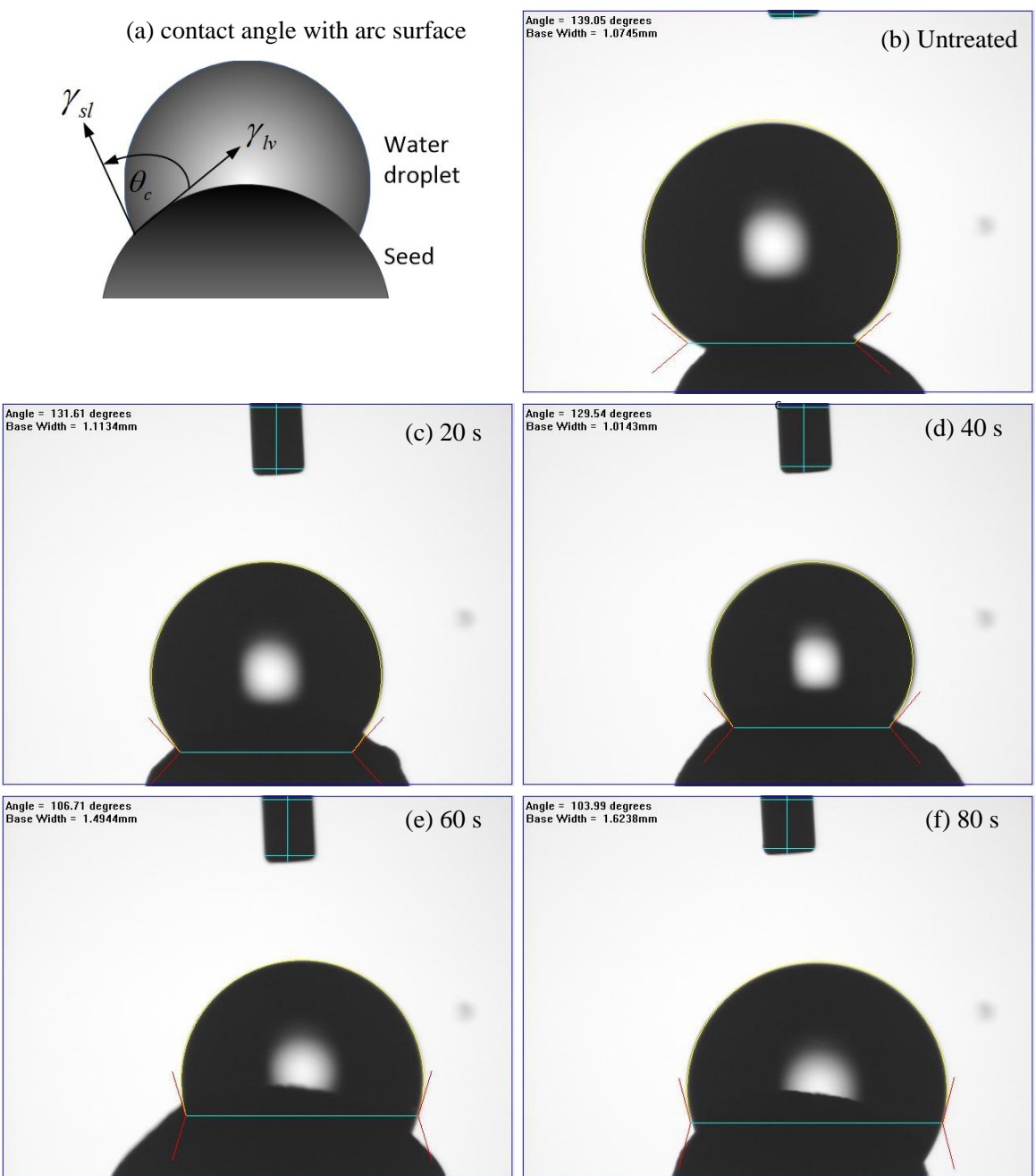

**Figure 6.** Contact angle measurement (**a**) schematic diagram of the contact angle with the arc surface, (**b**–**f**) contact angle images of the water droplet on broccoli seed with different treatment times.

### 3.3. SEM and XTM Images

Figure 8 shows the scanning electron images of the appearance of a particular structure on the treated and untreated seed surface. The broccoli seed's outer surface morphology seemed to be slightly modified to a smoother surface by the plasma treatment lasting for

80 s. The plasma electrons and ion bombardment of the seed coat's outer layer can reduce the volcano-like protuberances, making the seed surface appear more granulated [40,41].

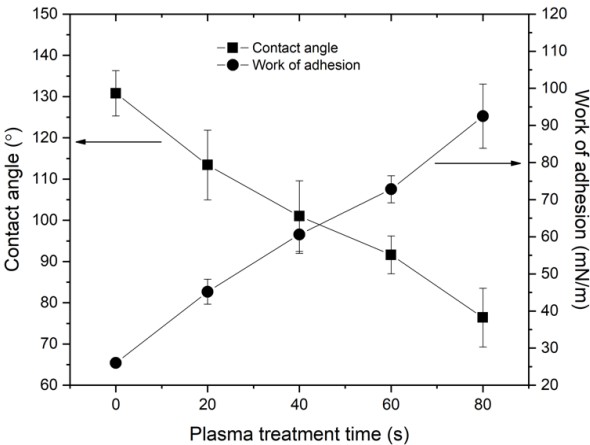

**Figure 7.** Contact angle and work of adhesion as a function of the plasma treatment time of broccoli seeds.

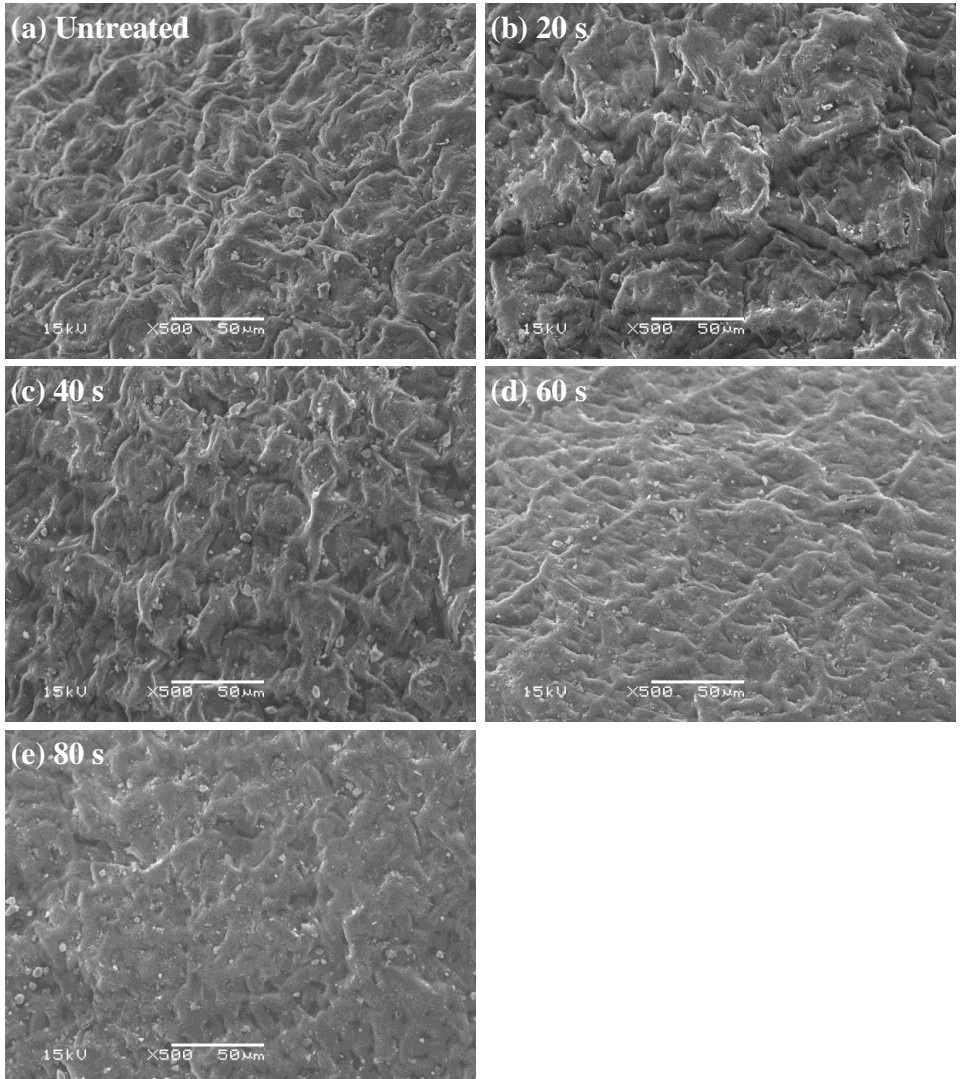

**Figure 8.** SEM image of broccoli seed (**a**) untreated, (**b**–**e**) plasma-treated for 20 s, 40 s, 60 s, and 80 s, respectively.

Figure 9 shows the XTM images of broccoli seed before and after plasma treatment. The whole seed's external surface is shown in Figure 9a,b, representing untreated and treated conditions. The XTM images show some rendering of the seed coat after being plasma treated for 80 s, resulting in a flatter surface when compared with the untreated sample. Figure 9c,d show the XTM images cross-sectional in the xz plane. This result reveals that the internal structure of the broccoli seed was not changed after the plasma treatment. In the plasma treatment process, the degree of ion energy may affect only the surface of the seed and cannot penetrate the seed coat and embryo. These results are in good agreement with the SEM analysis.

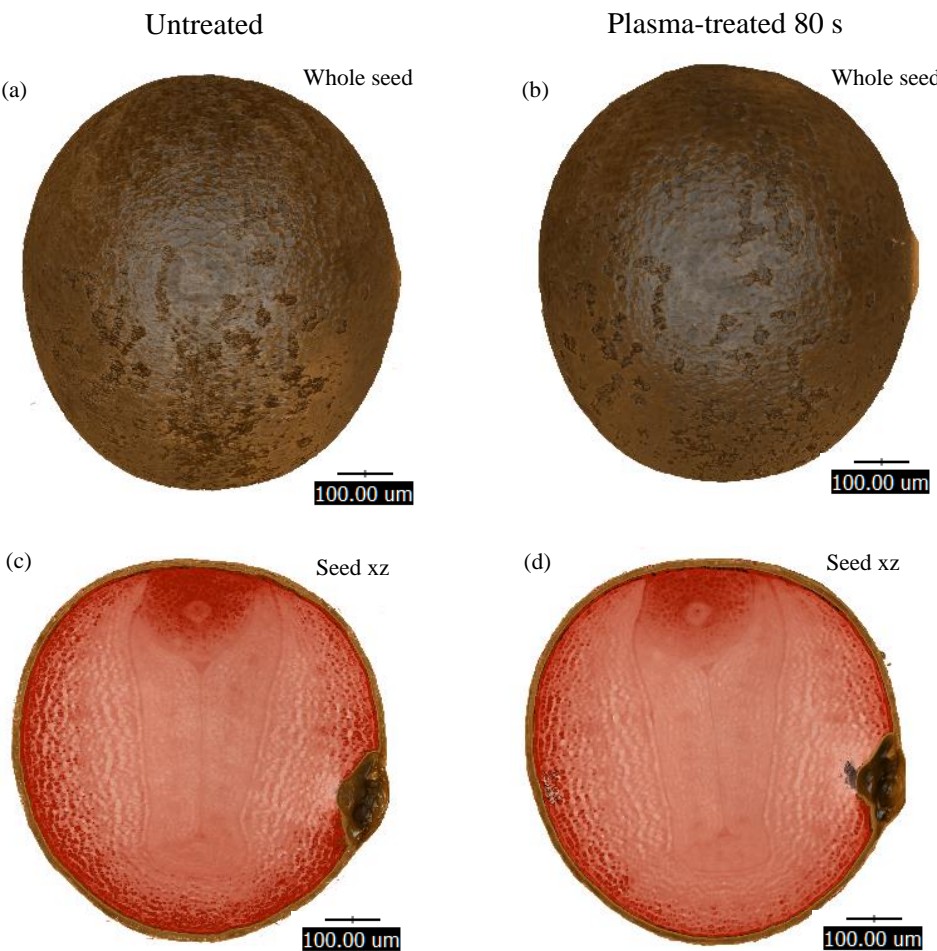

**Figure 9.** SRXTM image of (**a**) untreated, (**b**) plasma-treated, and cross-section in the xz direction of (**c**) untreated, and (**d**) plasma.

### 3.4. Effect of Plasma Treatment on Broccoli Seed Germination

Seeds treated by plasma showed an increased percentage of germination in comparison to the untreated seeds. Seven days after sowing, we found that the 30 s plasma-treated had the highest shoot length of $4.53 \pm 0.12$ cm, followed by seeds treated by plasma for 60 s ($4.47 \pm 0.15$ cm) and the untreated seeds ($4.39 \pm 0.11$ cm) as shown in Figure 10a. Figure 10b compares the germination percentage on days 1–7 of broccoli seed treated and untreated by plasma. Germination rates began to be stable from two days onwards. The results showed that seeds treated for 30 s had the highest germination rate of $94 \pm 1.6\%$, followed by the untreated seed ($92 \pm 0.9\%$). The germination percentage of seeds treated with plasma for 60 s decreased to $87 \pm 2.9\%$. This result could be due to the effect of ion bombardment of the seed during the plasma exposure.

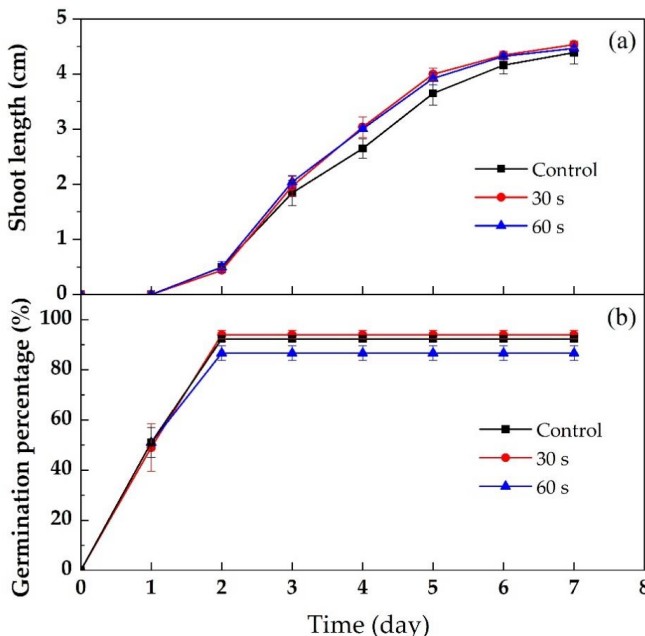

**Figure 10.** Effect of plasma seed treatment on broccoli cultivation (**a**) shoot length and (**b**) germination percentage on days 1–7.

Figure 11 compares the weight of seed, fresh sprouts, and additional weight of treated and untreated broccoli seeds after seven days of cultivation. The results showed that seed treated by plasma for 30 s obtained the highest yield. During plasma treatment, seeds were exposed to electrons, ions, UV, thermal radiation, and reactive species. Heat was the physical factor affecting the seed coats directly depending on the treatment time [32]. This phenomenon was attributed to the momentum transfer and chemical reactivity among radicals and ionic plasma species. Therefore, the increase in plasma treatment time can induce more reactive and energetic plasma active species that would negatively affect germination.

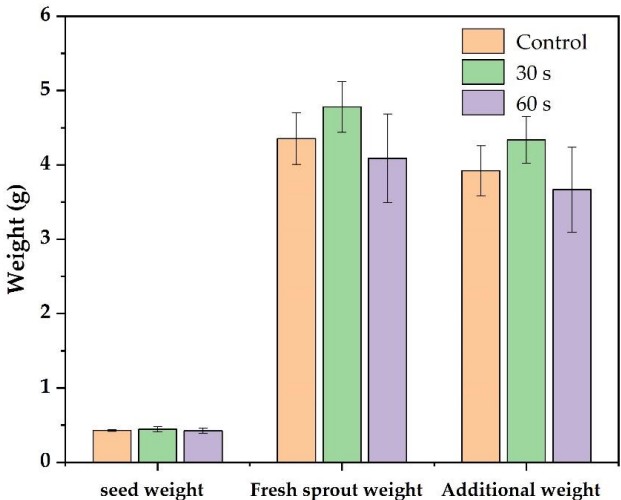

**Figure 11.** Comparison of the weight of sprouts of treated and untreated broccoli seeds after seven days of cultivation.

## 4. Conclusions

In this research, a multihole atmospheric pressure plasma jet with a scanning stage was developed to produce low-temperature plasma plumes for high-throughput large-

area treatment. The conditions that gave a stable plasma jet consisted of an argon flow rate of 4.2 lpm, plasma power of 412 W, and discharge frequency of 76 kHz. With 1-min plasma treatment, the plasma temperature did not exceed 27 °C. The surface morphology of seed coats seems to be slightly modified, while the SRXTM cross-sectional images show no significant difference between seeds untreated and seeds treated by plasma. The germination percentage of seeds treated with plasma for 30 s was $94 \pm 1.6\%$. This result is the optimum condition under the argon plasma treatment. Although the germination percentage was higher than that observed for the untreated seed, approximately 2%, a growth enhancement was also improved. After seven days of cultivation of treated broccoli seeds, the additional weight of sprouts was higher than that of untreated seeds by 10.5%. This result indicates that with a short treatment time, the multihole APPJ can modify the seed coats and shows a potential impact on the productivity of sprouts that could be useful in seed processing technologies.

**Author Contributions:** Each author participated sufficiently in the work to take public responsibility for appropriate portions of the content. Conceptualization, K.S. and A.C.; methodology and formal analysis, K.S., A.C., W.S., A.L., P.P., P.C., C.R., K.T. and S.T.; writing—original draft preparation and review and editing, K.S., A.C., W.S.; visualization, C.R., K.T. and S.T.; project administration, A.C. All authors have read and agreed to the published version of the manuscript.

**Funding:** This research was funded by the Electricity Generating Authority of Thailand, grant number 62-B602000-11-IO.SS03B3008434.

**Institutional Review Board Statement:** Not applicable.

**Informed Consent Statement:** Not applicable.

**Data Availability Statement:** All data are fully available.

**Acknowledgments:** The authors thank Adrian Plant for valuable suggestions and editorial comments on this manuscript.

**Conflicts of Interest:** The authors declare no conflict of interest. The funders had no role in the design of the study; in the collection, analyses, or interpretation of data; in the writing of the manuscript, or in the decision to publish the results.

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
