# Peer review of "Development of a Multihole Atmospheric Plasma Jet for Growth Rate Enhancement of Broccoli Seeds"

_processes, doi:10.3390/pr9071134_

Round 1

Reviewer 2 Report

Dear Authors,

the topic of your paper is very actual for last years with accent of germination of different seeds using different plasmas (especially atmospheric).

According to presented results and plasma acting, it will be very useful show results of chemical changes of treated surface (if is existed) depend of type of used gases, using FTIR or XPS equipment. Such results could confirm occurred changes as better wettability and maybe reason of better germination, or not. 

Also, could you explain changes of your observation and is it statistical significant difference from rate from 92% (untreated) to 94% (treated in optimal conditions)?

rows 208 - 212:  Germination rates began to be stable from two days 
onwards. The results showed that seeds treated for 30 s had the highest germination rate of 94%, followed by the untreated seed (92%). The germination percentage of seeds treated with plasma for 60 s decreased to 87%. This result could be due to the effect of ions bombardment of the seed during the plasma exposure.

General opinions of plasma technology (in textiles) which could be helpfull and cited in your paper:

....."The application of APPJs, which are characterized by flexibility and the unique property of external plume formation, is vastly growing.
Due to the pointed nature of the created discharge, it results in a localized small surface area of treatment. Additionally, interaction of the plasma plume with the surrounding atmosphere can have an effect on the physical
and chemical properties of the plasma. This can be solved by the application of a multi nozzle array in a controlled environment. It can be seen from the literature survey that there are many different approaches for achieving functional textiles with added value using plasma technology. Plasma is a versatile technique, primarily applied as a pretreatment process for cleaning,
etching, and activation of textile surfaces. The processes mentioned improve wettability of both natural and synthetic fibers. Chemical and physical etching of the surface can cause higher frictional forces between fibers, thus enhancing the breaking strength and adhesion properties. An etching effect can be applied in desizing where modification and degradation of the
sizing agent can significantly improve the desizing process. APP treatment has even been applied as a method for improving bioscouring and decoloration of indigo dyed denim for an aged-look effect. The plasma activation process imparts new functional groups, thus increasing affinity toward chemical agents used in textile finishing processes. Additionally, conducted research has proved that APP treatments have a potential
application in degumming of silk, as well as improving the piling, felting, and shrinking resistance of wool. The application of APP treatment for surface
modification of synthetic fibers is significant because most synthetic fibers are characterized by high crystallinity, non-polar hydrophobic character,
low surface energy, tendency to build up static electricity, pilling, and lower wearing comfort. Physiochemical changes of the surface induced by
plasma significantly improve the mentioned disadvantages of synthetic fibers, consequently improving their processability and finishing. A vast variety of systems developed for APP polymerization open up new possibilities for imparting much desired properties to textile substrates without degrading their inherent desirable properties. APP technology is adaptable and can be integrated into existing industrial lines for continuous
material processing. It is important to mention that plasma technology is being used in the textile industry not only for textile surface modification but
also as an advanced oxidation process for wastewater remediation".

Reference: Application of atmospheric pressure plasma technology for textile surface modification, Textile Research Journal, 2020, Vol. 90 (9–10) 1174–1197.

With best regards,

Reviewer

Reviewer 3 Report

This manuscript describes the use of multihole atmospheric plasma jet for the increasing of broccoli seeds growth. The manuscript is interesting, but it needs some ameliorations in the whole text.

Some points should be taken in consideration by the authors before the publication.

Starting from the introductions, the sentence that starts at lane 33 needs of some references. Moreover, can the authors give me some examples about the sentence at lane 38-40? How can ROS and RNS retain the quality of seeds and food production? It is also not clear why do you choose broccoli seeds. Have they some problem in germination? Why do the authors mention the sulforaphane amount in broccoli sprouts? They did not quantify it. Does this treatment influence the amount of sulforaphane in broccoli sprouts?

Materials and methods are clear and well written, and the experimental procedure is well described.

For the results and discussion section, I suggest adding some references in lane 126, can the authors make some examples? What kind of processes are produced by reactive oxygen and nitrogen species? Are the authors able to determinate them or justify them with some references? Figures 10 and 11 lack of standard error and statistical analyses. Are the authors sure that they can write at lane 209 “The results showed that seeds treated for 30 s had the highest germination rate of 94%, followed by the untreated seed (92%)”? 94 and 92 are statistically different? Moreover, could the authors give me some examples about the sentence at line 220?

Furthermore, I suggest the authors improving the conclusion part, it seems a sort of short summary of the previous part.

Round 2

Reviewer 2 Report

Dear Authors,

thank you for make overall changes and after such corrections paper will be very useful to scientific community and readers. I accept corrections.

We waiting for your next manuscript with results of plasma impact on chemical groups and surface reactivity which have potential influence to improve germination of broccoli seeds.

And, try to change and finally optimized some of parameters during your next experiment.

Looking forward to hearing you soon.

Best regards,

Reviewer